# Is Attention All You Need for EEG to Predict Neurological Outcomes in Cardiac Arrest?

Jefferson Dionisio [1], Muhammad Rizwan [2], Hidetaka Suzuki [3,4], Kazi Mahmudul Hassan [3],
Ingon Chanpornpakdi [3], Lian-Yu Lin [1,5], Toshihisa Tanaka [3], Che Lin [1,2,6,7,8*]

*Abstract*—Sudden cardiac arrest (SCA) poses a significant health challenge, necessitating accurate predictions of neurological outcomes in comatose patients, where good outcomes are defined as the recovery of most cognitive functions. Electroencephalogram (EEG) serves as a valuable biomarker for monitoring neurological states due to its rich, time-dependent information. This study aims to predict neurological outcomes using early EEG data by employing the Transformer model, which leverages multi-headed attention to identify patterns in lengthy sequences such as hour-long EEG recordings. Unlike traditional methods that use subsampled EEG epochs, we utilize the entire EEG sequences, subdivided into time steps, allowing our model to capture detailed temporal patterns via the attention mechanism. Moreover, we trained our proposed model using each EEG recording as an individual data sample but evaluated our model through aggregated patient-wise predictions. This approach allows us to boost the data sample size. Our results demonstrate promising predictive performance, achieving an AUROC of 0.82 and AUPRC of 0.90 on the holdout test set and an AUROC of 0.73 and AUPRC of 0.93 on an external test set with patient-wise predictions. This study highlights the potential of utilizing attention mechanisms to capture important time series progressions across EEG sequences for improving SCA prognosis.

*Index Terms*—cardiac arrest, EEG classification, multi-head attention, outcome prediction, time series data, Transformer

## I. INTRODUCTION

IN recent years, the prevalence of cardiac arrest has risen due to various factors such as unhealthy lifestyles and diets [1]. People who experience sudden cardiac arrest (SCA) often arrive in the intensive care units (ICU) of hospitals several minutes or hours after their initial time of cardiac arrest. Depending on this length of duration, such patients would often suffer from hypoxic-ischemic brain injury due to prolonged lack of oxygen [2], [3]. This, in turn, may result in patients remaining in a comatose state with uncertain future outcomes. Physicians are often asked to estimate patient prognosis during the first few days after the return of spontaneous circulation (ROSC). However, current clinical protocols suggest that physicians only deliver patient prognosis 72 hours after ROSC. Consequently, there has been significant interest in developing reliable predictive models to assist physicians in making crucial clinical decisions at this critical 72nd-hour juncture.

Electroencephalogram (EEG) is low-cost, non-invasive, and serves as an effective biomarker that aids clinicians in understanding brain conditions through visual inspection of brain waves. Recently, there has been a growing trend in releasing open-source EEG datasets for online challenges, such as the 2018 PhysioNet Challenge [4] and the 2003 BCI Competition [5], indicating an increasing interest in developing computational methods for analyzing EEG data.

In 2023, PhysioNet released its largest dataset to date through its annual George B. Moody PhysioNet Challenge [6], [7], with data consisting of EEG, electrocardiogram (ECG), and various clinical data from comatose SCA patients gathered from various hospitals across the US and Europe. The goal of the challenge was to build a model that can accurately predict the neurological outcomes of patients 3 to 6 months later, given early EEG and other data recorded hourly after the patients' ROSC. In this study, we aim to use the open dataset released from this challenge to train a time-dependent deep learning model that can effectively learn EEG recordings' progressions over time to predict patients' neurological outcomes.

Clinically interpretable features were extracted from each time step in the EEG sequence to help the model understand the EEG's progression and associate it with patient outcomes. Each EEG sequence serves as individual training data, enabling the model to learn recording-specific patterns and generalize to unseen data, regardless of the recording time after ROSC. We aggregate these recording-specific predictions to evaluate the model on a patient-wise basis. Finally, we use an external private dataset from National Taiwan University Hospital (NTUH) to test the model's generalizability further.

In recent years, the Transformer [8] has demonstrated exceptional performance with time series data due to its multi-head attention mechanism, which enables efficient learning from multiple time steps in a sequence and processing them in parallel with the multiple heads. The attention mechanism

"This work was supported in part by Taiwan's National Science and Technology Council, Ministry of Health and Welfare, and Ministry of Education under MOST 110-2221-E-002-112-MY3, MOHW112-TDU-B-221-124003, and NTU-112L900701, respectively."

[1] Department of Smart Medicine and Health Informatics, National Taiwan University, Taipei, Taiwan
[2] Artificial Intelligence of Things, Taiwan International Graduate Program, Academia Sinica, Taipei, Taiwan
[3] Tokyo University of Agriculture and Technology, Tokyo, Japan
[4] Department of Emergency and Critical Care, Japanese Red Cross Musashino Hospital, Tokyo, Japan
[5] Division of Cardiology, Department of Internal Medicine, National Taiwan University College of Medicine and Hospital, Taipei, Taiwan
[6] Department of Electrical Engineering, National Taiwan University, Taipei, Taiwan
[7] Graduate Institute of Communication Engineering, National Taiwan University, Taipei, Taiwan
[8] Center for Advanced Computing and Imaging in Biomedicine, National Taiwan University, Taipei, Taiwan
* Correspondence: chelin@ntu.edu.tw

in the Transformer allows the model to relate different parts of the sequence to each position, allowing it to learn long-distance relationships among each position. Incorporating positional encodings further enables the model to learn time-dependent patterns across various positions in a sequence. Given the highly time-dependent nature and length of continuous EEG sequences in our study, we hypothesize that the attention mechanism in the Transformer can work well with the continuous EEG and may deliver promising results. We propose a method for training the Transformer model on hourly EEG sequences segmented to form a time series dataset to fully utilize the attention mechanism of the model.

It is a common preprocessing strategy when dealing with EEG to subsample an epoch from a sequence of EEG. However, it is intuitive that we are wasting rich, valuable data when we only subsample from a long sequence of continuous EEG. To the best of our knowledge, this is the first study to use the full hours of EEG recordings from comatose cardiac arrest patients, subdivided into multiple time steps, to train a Transformer-based model to predict neurological outcomes. In this study, we test our hypothesis that carefully trained attention on full-length hourly EEG sequences is more effective than subsampling from each sequence for predicting neurological outcomes and aiding clinicians' patient prognosis.

## II. RELATED WORKS

Previous studies have used coma patients' EEG to predict patient neurological outcomes [9], [10]. Wennervirta et al. [9] gathered 30 coma SCA patients' EEG data from the ICU of Helsinki University Hospital and used statistical methods such as the chi-square test to make predictions using clinically interpretable features such as burst-suppression ratio, response entropy, state entropy, and wavelet subband entropy. Cloostermans et al. [10] gathered 56 coma SCA patients' EEG data from the ICU of Medisch Spectrum Twente, Enschede, The Netherlands. Similarly, using statistical methods, they built their predictive model using absent short-latency (N20) SSEP as the input feature. Both studies automatically selected 5-minute epochs from every hour of recording. They showed that the first 24 hours of EEG after ROSC already had good discriminative abilities to differentiate between good and bad outcomes.

With the advent of machine learning-based computing methods and their promise of better results, researchers often choose to transition from conventional statistical methods to machine learning and even deep learning methods [11]–[15]. A study [16] analyzed EEG data from 69 comatose SCA children, selecting the first artifact-free 5-minute epoch per hour from all available recordings. They extracted 8 quantitative features, including spectral density, normalized band power across five frequency bands (delta 0.5–3 Hz, theta 4–7 Hz, alpha 6–12 Hz, beta 13–30 Hz, gamma 25–50 Hz), line length, and regularity function scores. These features were used to train random forest, logistic regression, and support vector machine models for two setups: early EEG (0–17 hours post-ROSC) and late EEG (18 hours onward). The results indicated that early EEG had better predictive capability regarding accuracy, sensitivity, and specificity.

Another study [11] utilized the dataset from the 2023 PhysioNet Challenge to develop a bidirectional long short-term memory (bi-LSTM) model that learns long and short-term time dependencies. They used nine clinically interpretable features as input: burst suppression ratio, Shannon entropy, $\delta$ (0.5–4 Hz), $\theta$ (4–7 Hz), $\alpha$ (8—15 Hz), $\beta$ (16—31 Hz) band power, $\alpha/\delta$ ratio, regularity, and spike frequency. Bipolar referencing was employed to reduce channel-wise artifacts. Additionally, they scored the signal quality of each 5-minute epoch per hour of recording to be used as weights, aiding their model during training. Their model achieved an AUROC score of 0.78 at 12 hours and 0.88 at 66 hours, showing that performance improves over time with these features.

The Transformer model [8] has recently been applied to various time series data. Its efficiency in handling long-distance dependencies and learning patterns through parallel processing with multiple heads and positional encoding makes it well-suited for these applications. For instance, Wu et al. [17] have applied the Transformer to wind speed data and achieved promising wind speed forecasting results. Another study [18] used the Transformer for multimodal data, fusing doctors' clinical notes with structured EHR data, further indicating its adaptability to diverse datasets.

Among time-series models, there are several notable studies [19]–[21] that have compared the Transformer with traditional ones like LSTM and BLSTM. The aforementioned studies consistently showed the Transformer outperforming the traditional models on various relevant time series datasets such as ECoG and EEG.

When it comes to EEG, there has also been numerous research that leveraged the Transformer for classification tasks. For instance, Du et al. [12] utilized EEG data with the Transformer to develop a model for person identification. Yan et al. [13] used scalp EEG data with the Transformer for seizure prediction tasks. Guo et al. [14] used EEG with the Transformer for emotion recognition and visualization tasks, while Zeynali et al. [15] used it for motor imagery classification.

Randomly sampling an epoch from an EEG sequence, as commonly done in previous studies [16], [18], is effective. However, using full EEG sequences as raw data for training Transformer-based models, as demonstrated in several studies [12], [20], [22], avoids the potential waste of valuable biological data inherent in sampling only small portions.

In the 2023 George B. Moody PhysioNet Challenge, some studies [23], [24] used the Transformer to predict neurological outcomes with randomly selected 5-minute epochs but were not evaluated in the final challenge. Our previous study [25] achieved competitive results by using Transformers with features from the last hour of EEG recordings, incorporating both clinical and EEG data as model inputs.

## III. MATERIALS AND METHODS

### A. PhysioNet Dataset

For the 2023 George B. Moody PhysioNet Challenge [6], [7], the International Cardiac Arrest Research Consortium (I-CARE) [26] gathered comatose cardiac arrest patients' data from seven hospitals across the US and Europe. Overall, the dataset contains EEG data collected from 1020 patients, where 60% of the dataset was used as the training set, 10% as the hidden validation set, and 30% as the hidden test set. This study primarily utilized the entire publicly available training set to train and evaluate the model, dividing it into an 80% training set and a 20% test set.

The dataset's labels were based on outcomes and Cerebral Performance Category (CPC) scores obtained 3 to 6 months post-ROSC. CPC is a widely used 5-point scale assessing cognitive recovery: 5 for death, 4 for the persistent vegetative state, 3 for severe disability, 2 for moderate disability, and 1 for good recovery [27]. Labels were assigned as "good" (0) for CPC values of 1 and 2 and "bad" (1) for CPC values of 3, 4, and 5.

### B. NTUH Dataset

The NTUH dataset, a private dataset that was collected from National Taiwan University Hospital, was used as the external test set to evaluate the generalizability of our model. This dataset consists of 75 coma patients who had been resuscitated following cardiac arrest and had been comatose between 2013 and 2017 in the ICU of NTUH. 12 patients were defined as good outcomes, and 63 were defined as bad outcomes. The same EEG channels used in the PhysioNet dataset were used here to maintain consistency with the training dataset.

### C. Study Design

We used all recordings from the first 80% of the patients, 485 patients, from the publicly available training set released by PhysioNet to train our model and the recordings from the last 20% patients, 122 patients, to evaluate the model as the holdout test set. The resulting class distribution for the training set from this split is 39.38% for class 0 and 60.62% for class 1, which possesses a similar class distribution from the entire dataset with a ratio of 37.07% to 62.93%.

For this study, we aimed to focus on EEG. Thus, the other channel groups and the clinical data were not used. Given the superior predictive capabilities of early EEG [9], [10], [28]–[30] and the study's goal of aiding clinicians with patient prognosis at the 72-hour mark after ROSC [31], we used only EEG recordings taken within this timeframe.

The Transformer is a computationally expensive time series model; thus, to make use of the entire EEG sequences on this large dataset that consists of hourly recordings, we first extract clinically interpretable EEG features, particularly power spectral densities (PSD) among frequency band—delta (0.1 to 4.0 Hz), theta (4.0 to 8.0 Hz), alpha (8.0 to 12.0 Hz), and beta (12.0 to 30.0 Hz), to serve as the quantitative features for each time step in each EEG sequence. We opted to only use PSD features to maintain simplicity and consistency in our

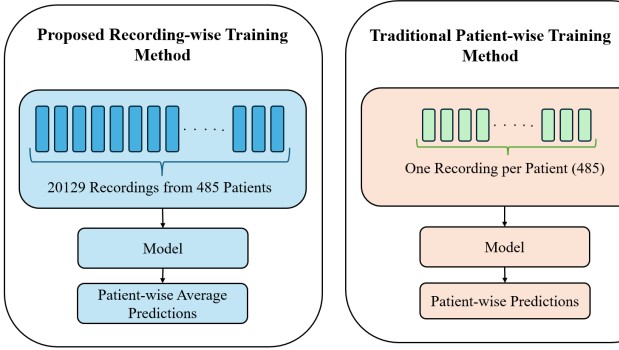

Fig. 1: Our proposed recording-wise training vs. traditional patient-wise training method

analysis, avoiding the noise and complexity that could result from combining different types of features as what previous studies did [9], [11], [16].

Our study slightly differs from previous studies on this dataset, particularly studies from the challenge, which primarily focused on patient-wise training using EEG with clinical data [25], and using ECG in addition to EEG [32]. Here, we focused only on utilizing EEG data to train our model. We trained our model using individual hours of EEG recordings as inputs rather than training the model on a patient-by-patient basis. For evaluation, we averaged the recording-wise predictions of each patient to obtain patient-wise predictions. This approach allows us to increase our sample size, enhancing model performance. The Transformer benefits significantly from larger datasets, as they can learn better attention weights with more data. Additionally, it helps the model learn recording-wise patterns, allowing it to generalize unseen data better, regardless of recording time. Figure 1 shows our proposed method of training our model recording-wise rather than the traditional way of patient-wise training.

Additionally, we utilized complete EEG sequences instead of using only randomly sampled 5-minute epochs as previous studies from the challenge did [23], [24]. We hypothesize that this approach would allow our model to learn more effective attention weights and make better predictions by capturing temporal patterns across entire continuous EEG recordings rather than local patterns within a subsampled epoch. Figure 2 shows a side-by-side comparison of our proposed method of using entire sequences with the traditional method of processing EEG using a subsampled epoch from the sequence.

### D. Data Preparation

A total of 20129 EEG recordings from 485 patients were used as training data. Each EEG sequence was first segmented into 5-minute epochs to serve as the time steps for the Transformer model. Bad epochs were automatically dropped from each EEG sequence by MNE [33], and the remaining good epochs were used and processed further. All 19 EEG channels from the good epochs were used, and bandpass filtering of [0.1, 30] Hz was used to filter out unwanted frequencies from the EEG data. The EEG sequences were all resampled from the

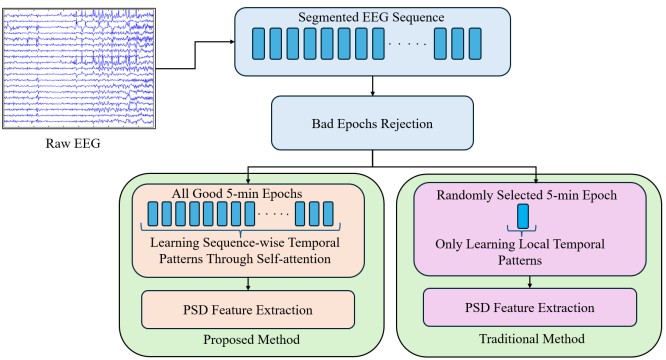

Fig. 2: Maximizing attention-wise learning by using entire sequences (lower left), compared to traditional random epoch selection (lower right).

original sampling rate of 500 Hz to 128 Hz, then normalized to values from $-1$ to $+1$.

After the signal processing steps, the mean PSD features were computed from each EEG sequence by first calculating the PSD for the delta (0.1 to 4.0 Hz), theta (4.0 to 8.0 Hz), alpha (8.0 to 12.0 Hz), and beta (12.0 to 30.0 Hz) frequencies from each channel. This was done by transforming the time samples to the frequency domain using fast Fourier transform with Welch's method [34], then obtaining the average power for each channel. The resulting mean PSD features for each channel were concatenated into a single feature vector, totaling 76 quantitative EEG features (19 EEG Channels $\times$ 4 Frequency Bands $\times$ 1 mean PSD = 76 features) from each 5-minute epoch in each EEG sequence. These features were subsequently used as the input for our model.

Each hour of EEG recording varied in length due to different start and end times. For shorter recordings, zero padding was added to ensure uniform length. If a recording started late, such as 20 minutes into the hour, the first four 5-minute epochs were padded with zeros. Similarly, if a recording ended early, zeros were added at the end to complete the hour. For recordings interrupted mid-hour and continued later, the segments were concatenated into one hour. Masks were created for each EEG sequence to help the model focus on actual data and ignore zero-padded segments. All EEG features were scaled from 0 to 1 to enhance convergence speed and numerical stability during model training. Both the PhysioNet test set and the NTUH dataset were preprocessed the same way as the training set.

*E. Model Training*

Our model architecture consists of the Transformer encoder [8] as part of the main block of the architecture, with the batch of EEG features used as the input. In the original implementation of the Transformer, given a sequence as input, the sequence is first divided into several tokens, each with a $d_{model}$ sized embedding. Here in this study, we treated each 5-minute epoch of an EEG sequence as tokens, and the extracted EEG features are used as the original embedding. Note that in each 5-minute token, the mean PSD features were used.

In this case, both temporal and frequency-domain features were included. These time series tokens enable the model to maximize its capability to learn temporal patterns across the EEG sequences through self-attention.

The Transformer encoder block consists of the multi-head attention (MHA) and feed-forward (FF) blocks. The input EEG features are initially encoded via a linear embedding layer into embeddings. Then, positional encoding, computed using sinusoidal functions (sine and cosine functions of different frequencies) as described in the original implementation [8], is added to each token in the EEG sequence to enable the model to learn the temporal dynamics among each token.

The MHA block in the Transformer includes a residual connection that adds the original input back to the output after MHA computation, followed by layer normalization. MHA is computed by first calculating three vectors: query ($Q$), key ($K$), and value ($V$). These vectors are derived from the input ($X$) through learned weight matrices. Specifically, the computations are: $Q = XW^Q, K = XW^K, V = XW^V$, where $W^Q, W^K, W^V$ are learned weight matrices. The scaled dot product attention is then computed as: *Attention*$(Q, K, V) =$ *Softmax*$(QK^T/\sqrt{d_k})V$, where $d_k$ is the dimension of the key vectors and serves to normalize the dot product of Q and K. Multihead attention extends this by projecting the queries, keys, and values into multiple subspaces (or heads) and performing the attention operation in parallel: $head_i =$ *Attention*$(QW_i^Q, KW_i^K, VW_i^V)$. Finally, the outputs of these parallel attention heads are concatenated and linearly transformed to produce the final output of the MHA block: *MHA*$(Q, K, V) =$ *Concat*$(head_1, \ldots, head_h)W^O$, where $W^O$ is another learned weight matrix. The computed masks from the data preparation step were used in the MHA block to help the model avoid learning from the zero-padded segments by assigning lower attention weights to the padded tokens. After the residual layer is added, layer normalization is applied, and the resulting embedding of the same dimension is then passed to the FF block.

In our model, the FF block consists of an expansion layer followed by ReLU activation and dropout and then a contraction back to the original dimension. A residual connection adds the embedding from before the FF block to help retain important information from the original data.

Global average pooling is then used to obtain the mean output from all time steps (tokens) in each sequence in the batch. Here, the masks are used again to remove the zero-padded tokens from being included in the global average pooling computation. The fully connected (FC) block consists of 3 contraction layers to eventually leave a single output for each EEG sequence. Finally, the sigmoid activation function transforms the outputs into probabilities.

During model training, prediction probabilities are generated for each recording from all patients. These probabilities are then averaged across the recordings of each patient to produce the final patient-wise predictions. The threshold for classifying these probabilities is optimized during training to achieve the best F1 score and accuracy. Instead of using the

TABLE I: Results from 80% PhysioNet training set

| Metric | PhysioNet Test Set | NTUH dataset |
|---|---|---|
| AUROC | 0.82 | 0.65 |
| AUPRC | 0.90 | 0.90 |
| Accuracy | 0.73 | 0.74 |
| F1 score | 0.79 | 0.84 |

default threshold of 0.5, the threshold is adjusted to improve performance on imbalanced datasets. This optimized threshold, which maximizes accuracy and F1 score, is then used consistently during subsequent model validation and testing.

*F. Experimental Setup*

We trained the model using cross-validation (CV) with k=5 across the entire training dataset and used the following metrics on the patient-wise predictions and prediction probabilities: AUROC, AUPRC, accuracy, and F1 score. Our criteria for choosing the model is through early stopping, defined as when the patient-wise results' AUPRC from validation has not improved beyond the current best for 10 consecutive epochs.

Our optimized model had the following hyperparameters, with the number of layers for the encoder as 3, the dropout rate at 0.4, 64 embedding size, 8 heads, batch size of 16, and a learning rate of 0.0001. To avoid overfitting, we introduced a decay rate of 0.99 after every 50 epochs and trained further until 200 epochs or early stopping criteria were reached.

## IV. RESULTS AND DISCUSSIONS

Using the chosen model during CV, we evaluated its performance on the PhysioNet test set (the last 20% of patient recordings from the public dataset) and the NTUH dataset. The optimized prediction threshold, determined during model training, was set at 0.55 and applied to both test sets, as shown in Table I. High metric scores were observed with the PhysioNet test set, with 0.82 AUROC, 0.90 AUPRC, 0.73 accuracy, and 0.79 F1 score, while the model achieved promising results on the NTUH dataset, with an AUROC of 0.65, an AUPRC of 0.90, an accuracy of 0.74, and an F1 score of 0.84. The poorer performance of the NTUH dataset is expected, as it serves as an external test set to evaluate the model's generalizability. External test sets often reveal the model's limitations in adapting to new, unseen data.

We evaluated our model's performance at specific time intervals after ROSC by evaluating it at 12, 24, 48, and 72 hours post-ROSC using the PhysioNet test set. We filtered the test set to include only EEG recordings within each specified time threshold. As shown in Figure 3, the model achieved its highest AUROC of 0.82 at 72 hours post-ROSC, while the lowest AUROC of 0.77 was observed at 24 hours.

These results are consistent with expectations, as our model was trained using EEG sequences for up to 72 hours. The superior performance at 72 hours highlights the model's potential as a clinical tool, aligning with the practice of making patient prognoses after 72 hours following ROSC. The model's best performance at this mark reinforces its reliability and suitability for clinical use in this critical time frame.

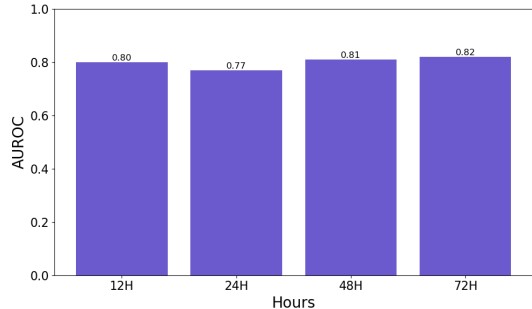

Fig. 3: AUROC from model evaluation at each hour threshold for the PhysioNet test set

Interestingly, the model performed better at 12 hours (AUROC of 0.80) than 24 hours (AUROC of 0.77). However, this slight decrease in performance is not very significant, highlighting the model's overall robustness even with earlier EEG recordings. This experiment confirms that the model has a comparable predictive capacity at earlier and later time points, with all AUROCs remaining above 0.77.

From these results, we can observe that the model demonstrates high performance from an early stage. This suggests that EEG data as early as 12 hours already provide clear and consistent signals for the model, which may help facilitate effective early risk stratification.

We further investigated how well our model performs on the different hospitals across the PhysioNet test set by showing the AUROC from each hospital in Figure 4. Here, Hospital E had the best AUROC at 0.85, while Hospital B had the worst at 0.33, which is way below the baseline of 0.5. Future studies may investigate removing some hospitals, such as Hospital B, or performing analyses on the different hospitals to better understand the reasons for the good and poor performances. It is also worth noting here that the second best AUROC was Hospital A at 0.84, which supports the data statistics since most of the hospitals from the training set came from Hospital A and may have led the model to learn more from the data of that hospital. The ratio of hospitals across the training set is as follows: Hospital A with 42.27%, Hospital B with 20.62%, Hospital D with 14.02%, Hospital E with 11.96%, and Hospital F with 11.13%. Each hospital was represented as alphabet letters in the dataset to protect patient privacies.

In this study, we utilized all EEG recordings individually to train the model, enabling it to learn recording-specific patterns and improved embeddings through a larger set of data samples. We conducted an experiment to compare the results of training our model with patient-wise (PW) data versus recording-wise (RW) data as input. For the PW setup, we used the last EEG recording per patient within 72 hours after ROSC, following the approach of a previous study from the challenge [25]. Table II presents a side-by-side comparison of results from both setups when evaluated with the PhysioNet test set and the NTUH dataset.

The results indicate a significant drop in AUROC and

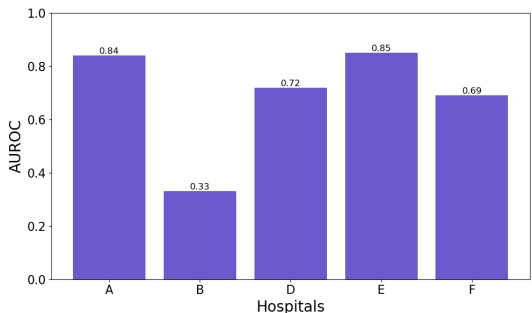

Fig. 4: AUROC from the model evaluation on each hospital for the PhysioNet test set

TABLE II: Comparison of RW and PW setups when evaluated with PhysioNet Test Set and NTUH Dataset

| Metric | PhysioNet RW | PhysioNet PW | NTUH RW | NTUH PW |
|---|---|---|---|---|
| AUROC | 0.82 | 0.58 | 0.65 | 0.61 |
| AUPRC | 0.90 | 0.76 | 0.90 | 0.89 |
| Accuracy | 0.73 | 0.71 | 0.74 | 0.78 |
| F1 score | 0.79 | 0.81 | 0.84 | 0.88 |

AUPRC when evaluated with the PhysioNet test set using PW data, highlighting the advantage of training with recording-wise data. Similarly, when evaluated with the NTUH dataset, the RW setup yielded better AUROC and AUPRC. These findings demonstrate the robustness of our proposed RW setup, as it performed exceptionally well on the holdout PhysioNet test set and generalized more effectively when evaluated with the NTUH dataset.

Since numerous studies on computing methods for EEG employed subsampling strategies by selecting a random 5-minute epoch to represent an EEG sequence, we use this strategy in an experiment to compare with our main setup. The main setup, labeled "Full Hours," utilizes full hourly EEG recordings, while the comparison setup, labeled "5-minute," uses only 5-minute segments to represent each hour of EEG. To maintain consistency in using the same model architecture as the main setup, we subdivided these 5-minute epochs into 10-second epochs as time steps for our model. No masking was necessary as only complete 5-minute epochs were selected, ensuring no missing data within the epochs.

Table III shows the results from both setups when evaluated with the PhysioNet test set and the NTUH dataset. We reprocessed the NTUH dataset for the comparison with the 5-minute setup to only use 1 randomly selected good 5-minute epoch for fair evaluations. Our findings show that the Full Hour setup heavily outperforms the 5-minute setup when evaluated with the PhysioNet test set. This supports our hypothesis that utilizing full hours of EEG instead of subsampling a portion of the EEG sequence shows better performance. The Transformer effectively leverages the rich and long time steps from EEG well through its attention mechanism to achieve good predictions.

TABLE III: Comparison of Full Hour and 5-minute setups when evaluated with PhysioNet Test Set and NTUH Dataset

| Metric | PhysioNet Full Hour | PhysioNet 5-minute | NTUH Full Hour | NTUH 5-minute |
|---|---|---|---|---|
| AUROC | 0.82 | 0.71 | 0.65 | 0.63 |
| AUPRC | 0.90 | 0.83 | 0.90 | 0.91 |
| Accuracy | 0.73 | 0.73 | 0.74 | 0.62 |
| F1 score | 0.79 | 0.80 | 0.84 | 0.75 |

TABLE IV: NTUH dataset results when trained with 80% training set compared to the entire PhysioNet dataset

| Metric | 80% Training Set | Entire Dataset |
|---|---|---|
| AUROC | 0.65 | 0.73 |
| AUPRC | 0.90 | 0.93 |
| Accuracy | 0.74 | 0.70 |
| F1 score | 0.84 | 0.80 |

When evaluated with the NTUH dataset, the Full Hour setup outperforms the 5-minute setup on almost all metrics except for the AUPRC, where the 5-minute setup obtained a score of 0.91 over 0.90 for Full Hour. This result may be because the NTUH dataset only contains short recordings that do not reach an hour, and almost all the recordings had to be zero-padded for the Full Hour setup. Nevertheless, our main model, Full Hour setup, could still generalize well to this external dataset despite the difference in the number of time samples available for each recording. One limitation of this method of utilizing full sequences of EEG is that oftentimes, hospitals are incapable of collecting numerous long sequences of hour-long recordings due to facility limitations and other clinical factors. However, our results in this comparison experiment show that our proposed model can still perform well despite being evaluated with shorter recordings.

Furthermore, to fully maximize the publicly available dataset from PhysioNet, we performed an experiment to train our model using the same setup but with the entire dataset and compared its performance when tested on the NTUH dataset with the model trained from the 80% training set. Table IV shows the results for both setups. We can observe a large improvement in the AUROC from 0.65 to 0.73 and a slight improvement in AUPRC from 0.90 to 0.93 when the model was trained with the entire dataset. Notably, the model trained with the entire dataset uses a different prediction threshold at 0.62 compared to the 0.55 threshold for the 80% split model.

We sought to understand how well the model trained with the entire dataset performed when evaluated with the NTUH dataset, so we further investigated the number of true positives, true negatives, false positives, and false negatives among the predictions. Our findings show that among the positive samples (bad outcome patients), it correctly predicted 45 (true positives) and incorrectly predicted 18 (false positives), while among the negative samples (good outcome patients), it correctly predicted 8 (true negatives) and incorrectly predicted 4 (false negatives). These results show that the model obtained an accuracy of 0.71 from the positives and 0.67 from the

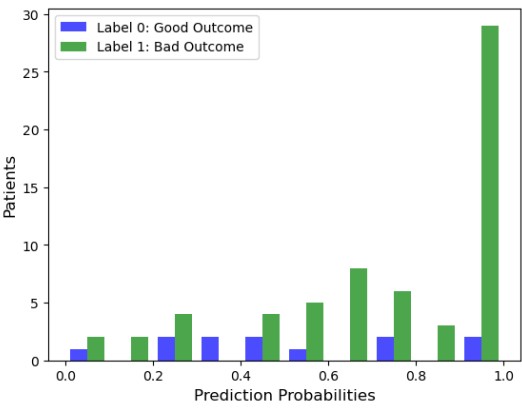

Fig. 5: NTUH dataset: prediction probabilities vs. true labels for the model trained on the full PhysioNet dataset

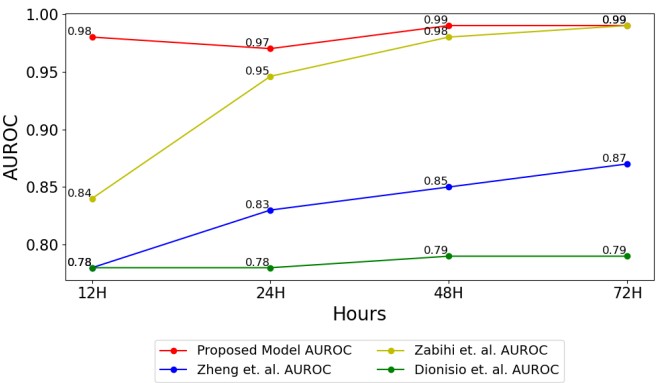

Fig. 6: AUROC benchmark comparison across the entire PhysioNet public training set via cross-validation

negatives, indicating the model's capability to distinguish well between the two classes despite the evident class imbalance.

Figure 5 shows the distribution between the true labels and the prediction probabilities generated by the model. The green bars represent the true label 1s (bad outcomes), and the blue bars represent the true label 0s (good outcomes). We can observe that a large portion of the green bars are beyond the optimized prediction threshold of 0.62, while the blue bars are mostly below the threshold, further demonstrating the model's excellent class separability. Overall, our model achieved very promising results on the external dataset when trained with the entire public dataset from PhysioNet, showcasing possible future applications in the clinical setting.

Finally, we evaluated our proposed model across the entire training set through CV and compared the resulting AUROC with previous studies, including those from the challenge, to be used as benchmark comparisons. We obtained the results from Zabihi et al.'s model [32] and our previous study's model [25] from the official results posted in the PhysioNet Challenge. Among the participants in the challenge who used attention-based models, only the model from our previous study was evaluated. Thus, the others were not included in this comparative analysis. We compared our model with Zabihi et al.'s despite their model not being time-dependent since they were the challenge winners, and it is vital to show where our proposed model stands compared to theirs. It is important to note that we cannot directly compare our findings with Zheng et al.'s results [11] since their reported results are from using CV across the entire dataset, including the two hidden validation and test sets. However, we included their results here to show how our model compares despite the difference in the data split. It must also be noted that only Zheng et al. used EEG as the sole input, while Zabihi et al. used ECG with EEG, and our previous study used clinical data with EEG to train the models.

We can observe from Figure 6 that our proposed model outperforms both Dionisio et al. (our previous study) and Zheng et al.'s models when evaluated at different hours. Inter-

estingly, our model had a very high AUROC when evaluated at 12H compared to the other benchmarks, which all have the worst performance at 12H. Overall, our proposed model outperformed the other benchmarks at all earlier time points and performed on par with the challenge winner's model when evaluated at 72H, despite being trained with only EEG data.

We note here that this benchmark comparison is only meant to show how our proposed model's results stand with the previous studies. Readers must take precautions when making direct comparisons since each benchmark's setups differ. However, all studies cited in this benchmark comparison have the same focus on predicting neurological outcomes from coma patients and make use of the same dataset.

This study has a few additional limitations beyond those previously mentioned. First, it focused exclusively on EEG data due to the study design. Future research could explore multimodal approaches to improve the methodology. Second, the emphasis was on deep learning techniques, and future studies may benefit from incorporating more advanced signal processing and channel selection methods. Lastly, exploring different types of EEG features in future studies could lead to further improvements.

## V. CONCLUSION

Our study's findings support our hypothesis that training the Transformer with full-hour lengths of EEG sequences is beneficial instead of subsampling and that the more data we use to train the model, the better the performance. We showed promising results with our proposed attention-based model through various analyses with an external dataset and various comparison metrics with previous benchmarks. We believe these findings show promising insights in trying to understand whether attention is all you need for EEG sequences to accurately predict neurological outcomes in comatose cardiac arrest patients. We hope to aid physicians in making important clinical decisions since our model can achieve highly competitive results using an attention-based model over continuous long EEG sequences.

ETHICS STATEMENT

The private dataset used in this study was approved by the National Taiwan University Hospital Ethics Committee.

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
