# OpenReview forum: "Is Attention All You Need for EEG to Predict Neurological Outcomes in Cardiac Arrest?"
_IEEE.org/EMBS/BHI/2024/Conference — IEEE BHI'24_

### Official Review · Reviewer_QecX · 2024-08-05
**Review points out weaknesses/limitations of the approach that are not discussed clearly/omitted from the paper. Despite limitations, the study idea/hypothesis, execution, and findings are interesting and important enough to warrant acceptance.**

**Overall Rating:** 7
**Confidence:** 4

**Other Quality Metrics:**

- (a) Clarity of writing - great
- (b) Clinical Significance - good
- (c) Methodological Novelty - fair
- (d) Experiments and Results - good

**Questions For The Authors:**

- Q: What EEG reference is used for the study? Is performance affected by referencing scheme (bipolar vs common average vs common reference, ...)? Does the external dataset use the same reference?
- Q: why was 5-fold cross-validation used on the 80% training set? Was it to 1) select optimal hyperparameters, or 2) to get one-final-model-per-fold, or 3) select a fold/subset of training data that gave the best metrics?
- Q: what exactly is the transformer loss function used for training? How is the threshold modified/optimized to reflect the best F1/accuracy metrics?
- Q: Is AUPRC providing additional value/insight beyond the AUROC as a performance metric (for example due to severe class imbalance)? Can you provide a random baseline score for the AUPRC metric analogous to the 0.5 AUROC random baseline for context?
- Q: Can you discuss the clinical workflow/healthcare system downstream implications of good/bad decision or accurate/inaccurate predictions at the 72-hour juncture? Does the asymmetric cost of false positives vs false negatives inform the choice of metric or threshold used for model evaluation or training objectives?

**Strengths:**

- The study presents compelling early evidence to the clinical EEG analytics community that there exists some learnable patterns in the long-range temporal EEG sequence and that using shorter EEG epochs is very likely a sub-optimal modeling choice.
- Study details how transformer-type models can be technically adapted to accept EEG signals as input, or other physiological signals more broadly.

**Summary Of The Paper:**

Study tests the clinical viability of using the transformer architecture on long EEG recordings with appropriate technical simplifications and modifications. The clinical application chosen is that of predicting neurological recovery of post-cardiac arrest comatose patients at the 72-hour time horizon.

**Weaknesses:**

- I would suggest making the paper title somewhat less provocative/grandiose given the instability of empirical performance across seen/unseen sites and the simplified clinical complexity of the prognosis task.
- The paper does not state the limitations of the proposed approach and potential future directions in discussion/conclusion sections.
- The explanation for Figure 5 sounds hand-wavy/subjective, it might help to redesign the plot or rephrase the observations to communicate the findings more objectively.
- Q: What can intuitively or technically explain the performance differences seen across sites (Figure 4) relative to the training data proportions? Sites with low (/high) support have disproportionately higher (/lower) performance. It may be interesting to plot mean PSD feature distributions of the training data per site and check if the amount of pairwise distribution overlap correlates with the magnitude of performance difference.
- Q: What explains the sharp drop in AH AUROC performance in Tables 2 and 3? Is it simply due to class imbalance (and therefore AUPRC is a better metric to check) or does it indicate severe site-related/dataset-related overfitting of the transformer model?
- Q: were clinical factors that systematically affect/bias the spectral/temporal profile of EEG controlled like age, medications, or presence of brain lesions? Is it possible that these factors are strongly correlated with the neurological prognosis in this clinical setting and need to be controlled for in the experiment design?
- Q: Was there any post-hoc interpretability investigation done to check if the model has learnt to pick up the same features as clinicians/those used in prior work? If not, what sort of short-range or long-range temporal patterns are being picked up by the model/attention heads? What features or characteristics of the EEG do clinicians rely on to estimate the prognosis? This would provide more context of the problem and the proposed solution and may be a direction for future research.

---

### Official Review · Reviewer_Ycjg · 2024-08-09
**Good experiments and benchmarks, pleasure to read**

**Overall Rating:** 7
**Confidence:** 5

**Other Quality Metrics:**

Clarity of writing Excellent
clinical significance Great
methodological novelty Good
experiment and results Excellent

**Questions For The Authors:**

When the record-wise predictions are pooled, how do you generate a patient-wise prediction? Is it mode of the prediction? It would be better to clarify it in the paper.

**Strengths:**

The paper is very well written and a pleasure to read. Though the modeling itself might not be as novel, the external validation of the proposed modeling is a great strength of the work. The methods are well described, and comparison with baselines is good (e.g., using full hour vs only some segments) and appreciated.

**Summary Of The Paper:**

The paper investigates EEG-based clinical outcome prediction using the Physionet challenge dataset. The paper proposes a transformer model utilizing the full hour of the EEG dataset for prediction. A record-wise prediction is proposed with late-pooling for patient-wise prediction. The modeling is independently validated with an external test set.

**Weaknesses:**

No major weakness.

---

### Official Review · Reviewer_9eCx · 2024-08-13
**Review on "Is Attention All You Need for EEG to Predict Neurological Outcomes in Cardiac Arrest?"**

**Overall Rating:** 5
**Confidence:** 5

**Other Quality Metrics:**

(a) Clarity of writing: good (b) Clinical Significance: good (c) Methodological Novelty: fair (d) Experiments and Results : good

**Questions For The Authors:**

It is suggested to justify why the use of a Transformer in the proposed study allows achieving good performances.

**Strengths:**

The use of Transformer to predict neurological outcomes in cardiac arrest. The proposed study has shown that training a Transformer with full-hour lengths of EEG sequences is beneficial instead of subsampling and that the more data are used to train the model, allows achieving better the performances.

**Summary Of The Paper:**

This paper studies the prediction of neurological outcomes using early EEG data by employing a Transformer model, which leverages multi-headed attention to identify patterns in lengthy sequences such as hour-long EEG recordings.

**Weaknesses:**

The performance of the use of a Transformer to predict neurological outcomes in cardiac arrest is not well highlighted nor justified.

---

### Decision · Program_Chairs · 2024-09-23

Accept